# Effect of internet-delivered cognitive behavioral therapy on insomnia in convalescent patients with COVID-19: Protocol for a systematic review and meta-analysis

**Yan Chen**[1], **Xiang Zhou**[2], **Jie Liu**[3], **Rui Li**[1], **Zihan Jiang**[1], **Lina Wang**[3], **Shuya Wang**[3], **Bo Li**[1] *

**1** Institute of Chronic Disease Risks Assessment, School of Nursing and Health, Henan University, Kaifeng, Henan, China, **2** Henan University of Traditional Chinese Medicine, Zhengzhou, Henan, China, **3** Huaihe Hospital of Henan University, Kaifeng, Henan, China

* 10210022@vip.henu.edu.cn

**Data Availability Statement:** Deidentified research data will be made publicly available when the study is completed and published.

## Abstract

### Introduction

Coronavirus Disease 2019 (COVID-19) has made a serious public health threat worldwide. Recent evidence has indicated that COVID-19 patients in convalescence frequently experience insomnia, which reduces their quality of life and causes unknown risks. The positive effect of cognitive behavior on insomnia has been well addressed in previous studies. Given the high infectivity and epidemicity of COVID-19, Internet-delivered intervention may be safer than face-to-face treatment. However, whether Internet-delivered cognitive behavioral therapy can effectively improve the insomnia of COVID-19 patients in convalescence has not been completely determined yet. Therefore, we conducted a meta-analysis and systematic review to evaluate the effects of Internet-delivered cognitive behavioral therapy on insomnia in COVID-19 patients in convalescence, with the aim to confer some guidance for its clinical application.

### Methods and analysis

This systematic review and meta-analysis has been registered in the International Prospective Register of Systematic Reviews (PROSPERO). Two researchers will retrieve the relevant literature on Internet-delivered cognitive behavioral therapy for insomnia in convalescent patients with COVID-19 in PubMed, Web of Science, Embase, MEDLINE, Cochrane Library, Clinical Trials gov, Chinese Biomedical Literature Database (CBM), and Chinese National Knowledge Infrastructure (CNKI) from inception to 11th of December. In addition, we will review the relevant trials and references of the included literature and manually searched the grey literature. The two researchers will independently extracted data and information and evaluated the quality of the included literature. The Review Manager software (version 5.3) and Stata software (version 14.0) will be used for data analysis. The

**Funding:** The authors received no specific funding for this work.

**Competing interests:** The authors have declared that no competing interests exist.

mean difference or the standardized mean difference of 95% *CI* will be used to calculate continuous variables to synthesize the data. In addition, $I^2$ and Cochrane will be used for heterogeneity assessment.

### Trial registration

PROSPERO registration number CRD42021271278.

## Introduction

World Health Organization (WHO) has reported that the ongoing pandemic of coronavirus disease 2019 (COVID-19) caused by the severe acute respiratory syndrome coronavirus 2 (SARS-CoV-2) has infected 200,840,180 people and resulted in 4,265,903 deaths worldwide by 6th August 2021 [1]. COVID-19 has rapidly developed into a pandemic and seriously threatened public health all over the world [2, 3]. COVID-19 pandemic seriously jeopardizes human physical and mental health [4] and attracts the concerns of numerous researchers [5]. The exploration of novel prevention, diagnosis, and treatment strategies has become a hot topic among scholars [6, 7], but little attention is paid to the mental health of COVID-19 patients in convalescence [8]. Studies have shown that the confirmed patients with COVID-19 tend to experience various psychological disorders such as stress, anxiety, depression, irritability, depression, and despair [9, 10]. Even though the mood of patients recovering from COVID-19 infection has improved, their symptoms of insomnia and fatigue still persist [11]. A considerable number of patients still suffer from insomnia and other symptoms of sleep disorders in the convalescent period [12]. A prospective cohort and nested case-control study has revealed that 29.17% of COVID-19 patients in convalescence experience insomnia [13].

Insomnia is defined as difficulties in falling asleep, maintaining asleep, or waking up too early at least three times per week for at least 3 months [14] and is one of the most common problems experienced by convalescent patients with COVID-19 [13]. COVID-19-related worries and loneliness are the major contributors to clinical insomnia in addition to education status, virus infection, and pre-existing mental health disorders [14, 15]. Poor sleep quality leads to adverse outcomes of physical [16] and mental health [17]. If left untreated, insomnia can aggravate anxiety, depression, and post-traumatic stress disorder and eventually compromise the quality of life [18]. Moreover, severe insomnia disorder or irregular sleep-wake cycle may damage the immune system and inflammatory response and increase the susceptibility to virus infection [19].

At present, primary health care and outpatient mental health institutions are responsible for the treatment of insomnia [20]. Drug therapy remains the most commonly used intervention method for insomnia. Despite the rapid curative effects, drug therapy fails to achieve radical treatment and also results in many adverse reactions [21–23]. Cognitive behavioral therapy, as an evidence-based option, has been found to effectively improve sleep outcomes [24, 25] and its therapeutic effect is equivalent to or better than that of drug therapy [26]. Since SARS-CoV-2 spreads through close contact between people [27], keeping a distance of more than two meters and using an N-95 mask are the most effective ways to prevent the spread of COVID-19 [28]. Under the guidance of therapists, the effect of Internet-delivered cognitive behavioral therapy on some mental and physical diseases is equivalent to that of face-to-face cognitive behavioral therapy [29], because the possibility of novel coronavirus pneumonia reinfection is unknown [30]. As a kind of digital therapy, Internet-delivered cognitive

behavioral therapy provides patients with evidence-based treatment interventions, which are driven by software to prevent, manage, or treat medical disorders or diseases [31]. It can be used independently or in concert with medications, devices, or other therapies [31]. Internet-delivered cognitive behavioral therapy represents a promising option to solve the obstacles of psychotherapy [32], which overcomes several limits related to traditional clinical practice [33] and possesses the advantages of low cost [31], high efficiency [31], and continuous monitoring [33]. Therefore, the cognitive behavioral therapy provided by the Internet may also have potential advantages in improving the insomnia of COVID-19 patients in the convalescent period.

The existing studies have shown that Internet-delivered cognitive behavioral therapy can improve symptoms of insomnia in adults [34]. However, a recent study has also found that Internet-delivered cognitive therapy has no significant effect on improving the insomnia of COVID-19 patients [35]. Therefore, this protocol is established to determine the effectiveness of Internet-delivered cognitive behavioral therapy for relieving insomnia symptoms in newly recovered patients with COVID-19.

## Review questions

Under the current COVID-19 epidemic background, can Internet-delivered cognitive therapy improve insomnia in rehabilitative patients?

## Objectives

The main purpose of this study is to conduct a systematic review and meta-analysis through randomized controlled trials (RCTs) to determine whether the Internet-delivered cognitive behavioral therapy can relieve insomnia in convalescent patients with COVID-19, improve the sleep quality of convalescent patients with COVID-19, and then improve the psychological condition of these patients.

# Materials and methods

The protocol is in line with the Preferred Reporting Items for Systematic Reviews and Meta-analysis for Protocol (PRISMA-P) [36].

## Inclusion criteria of the study

This systematic review protocol includes the following studies:

### Types of studies

We included randomized controlled trials (RCTs). These trials report the effects of Internet-delivered cognitive behavioral therapy on insomnia in convalescent patients with COVID-19.

### Participants

Participants are COVID-19 patients with sleep disorders in the convalescent period. Mental disorders will be involved in the meta-analysis.

### Ethics and dissemination

Since this study reviews the published data, ethical approval is not required. The results of this systematic review and meta-analysis will be published in peer-reviewed journals.

### Interventions

The intervention group will receive Internet-delivered cognitive behavioral therapy. Internet-delivered cognitive behavioral therapy will be conducted under the guidance of experienced psychotherapists or trained therapists. Patients will log into a secure website to access, read, and download online materials organized in a series of lessons or modules, including cognitive reconstruction, loosening practice, knowledge explanation, health education, and so on.

### Comparisons

Patients in the control group will receive routine care, such as self-regulation, medical consultation, and support.

### Outcome indicators

The sleep indicators for COVID-19 patients in convalescence will be as follows: (1) sleep quality (SQ), (2) time in bed, (3) sleep time (ST), (4) sleep efficiency (SE), (5) sleep disturbance (SD), (6) daytime function (DF), (7) Pittsburgh Sleep Quality Index (PSQI), and (8) Insomnia Severity Index (ISI).

### Search strategy

Two researchers (YC, XZ) will retrieve the relevant English and Chinese literature independently in the following databases from inception to 11th of December: PubMed, Web of Science, Embase, MEDLINE, Cochrane Library, ClinicalTrials.gov, Chinese Biomedical Literature Database (CBM), and Chinese National Knowledge Infrastructure (CNKI). PubMed literature search is as follows: (Table 1).

## Literature selection and data extraction

### Literature selection

The literature screening process will follow the steps in the PRISMA flow chart (Fig 1). Firstly, after retrieving all databases, we will use Endnote 9.0 software to eliminate duplicate studies. Then, the two researchers will independently examined the titles and abstracts of the documents obtained after weight removal and preliminarily screened the documents. Subsequently, the two researchers will independently read the preliminarily screened literature according to the inclusion and exclusion criteria we formulate in advance. Finally, the literature screened by the two researchers will be compared to determine the literature finally included in the analysis. A third independent researcher will be consulted to achieve a final consensus in case of disagreement between both researchers.

### Data extraction

The extraction of data and information in the literature will be completed by two researchers independently, including first author, publication date, country, intervention implementer, sample number, sample age, sample source, intervention time of online cognitive behavioral therapy, intervention frequency, dropout rate, insomnia assessment tool, and other details. Data will be collected and analyzed in the near future.

### Risk of bias assessment in the included literature

The two researchers will use the Cochrane bias risk assessment tool to independently assess the risk of publication bias of each included literature (random sequence generation,

**Table 1. Search strategy for PubMed.**

| Search number | Search term |
|---|---|
| #1 | (((((Internet[Title/Abstract]) OR (web[Title/Abstract])) OR (web-based[Title/Abstract])) OR (computer[Title/Abstract])) OR (online[Title/Abstract])) OR (digital[Title/Abstract]) |
| #2 | "Cognitive Behavioral Therapy"[Mesh] |
| #3 | ((((((((((((((((((((((Behavioral Therapies, Cognitive[Title/Abstract]) OR (Behavioral Therapy, Cognitive[Title/Abstract])) OR (Cognitive Behavioral Therapies[Title/Abstract])) OR (Therapies, Cognitive Behavioral[Title/Abstract])) OR (Therapy, Cognitive Behavioral[Title/Abstract])) OR (Therapy, Cognitive Behavior[Title/Abstract])) OR (Cognitive Behavior Therapy[Title/Abstract])) OR (Cognitive Therapy[Title/Abstract])) OR (Behavior Therapy, Cognitive[Title/Abstract])) OR (Behavior Therapies, Cognitive[Title/Abstract])) OR (Cognitive Behavior Therapies [Title/Abstract]))OR (Therapies, Cognitive Behavior[Title/Abstract])) OR (Cognitive Psychotherapy[Title/Abstract])) OR (Cognitive Psychotherapies[Title/Abstract])) OR (Psychotherapies, Cognitive[Title/Abstract])) OR (Psychotherapy, Cognitive[Title/Abstract])) OR (Therapy, Cognitive[Title/Abstract])) OR (Cognitive Therapies[Title/Abstract])) OR (Therapies, Cognitive[Title/Abstract])) OR (Cognition Therapy [Title/Abstract])) OR (Therapy, Cognition [Title/Abstract])) OR (Cognition Therapies[Title/Abstract])) OR (Therapies, Cognition[Title/Abstract]) |
| #4 | #2 OR #3 |
| #5 | "Sleep Initiation and Maintenance Disorders"[Mesh] |
| #6 | ((((((((((((((((((((((((((Disorders of Initiating and Maintaining Sleep[Title/Abstract]) OR (DIMS[Title/Abstract])) OR (Early Awakening [Title/Abstract])) OR (Awakening, Early[Title/Abstract])) OR (Nonorganic Insomnia[Title/Abstract])) OR (Insomnia, Nonorganic[Title/Abstract])) OR (Primary Insomnia[Title/Abstract])) OR (Insomnia, Primary[Title/Abstract])) OR (Transient Insomnia[Title/Abstract])) OR (Insomnia, Transient [Title/Abstract])) OR (Rebound Insomnia[Title/Abstract])) OR (Insomnia, Rebound[Title/Abstract])) OR (Secondary Insomnia [Title/Abstract])) OR (Insomnia, Secondary[Title/Abstract])) OR (Sleep Initiation Dysfunction[Title/Abstract])) OR (Dysfunction, Sleep Initiation[Title/Abstract])) OR (Dysfunctions, Sleep Initiation[Title/Abstract])) OR (Sleep Initiation Dysfunctions[Title/Abstract])) OR (Sleeplessness[Title/Abstract])) OR (Insomnia Disorder[Title/Abstract])) OR (Insomnia Disorders[Title/Abstract])) OR (Insomnia[Title/Abstract])) OR (Insomnias[Title/Abstract])) OR (Chronic Insomnia[Title/Abstract])) OR (Insomnia, Chronic[Title/Abstract])) OR (Psychophysiological Insomnia[Title/Abstract])) OR (Insomnia, Psychophysiological[Title/Abstract]) |
| #7 | #5 OR #6 |
| #8 | "COVID-19"[Mesh] |
| #9 | ((((((((((((((((((((((((((((((((((COVID 19[Title/Abstract]) OR (COVID-19 Virus Disease[Title/Abstract])) OR (COVID 19 Virus Disease[Title/Abstract])) OR (COVID-19 Virus Diseases[Title/Abstract])) OR (Disease, COVID-19 Virus[Title/Abstract])) OR (Virus Disease, COVID-19[Title/Abstract])) OR (COVID-19 Virus Infection[Title/Abstract])) OR (COVID 19 Virus Infection[Title/Abstract])) OR (COVID-19 Virus Infections[Title/Abstract])) OR (Infection, COVID-19 Virus[Title/Abstract])) OR (Virus Infection, COVID-19[Title/Abstract])) OR (2019-nCoV Infection[Title/Abstract])) OR (2019 nCoV Infection[Title/Abstract])) OR (2019-nCoV Infections[Title/Abstract])) OR (Infection, 2019-nCoV[Title/Abstract])) OR (Coronavirus Disease-19[Title/Abstract])) OR (Coronavirus Disease 19[Title/Abstract])) OR (2019 Novel Coronavirus Disease[Title/Abstract])) OR (2019 Novel Coronavirus Infection[Title/Abstract])) OR (2019-nCoV Disease[Title/Abstract])) OR (2019 nCoV Disease[Title/Abstract])) OR (2019-nCoV Diseases[Title/Abstract])) OR (Disease, 2019-nCoV[Title/Abstract])) OR (COVID19[Title/Abstract])) OR (Coronavirus Disease 2019 [Title/Abstract])) OR (Disease 2019, Coronavirus[Title/Abstract])) OR (SARS Coronavirus 2 Infection[Title/Abstract])) OR (SARS-CoV-2 Infection [Title/Abstract])) OR (Infection, SARS-CoV-2[Title/Abstract])) OR (SARS CoV 2 Infection[Title/Abstract])) OR (SARS-CoV-2 Infections[Title/Abstract])) OR (COVID-19 Pandemic[Title/Abstract])) OR (COVID 19 Pandemic[Title/Abstract])) OR (COVID-19 Pandemics[Title/Abstract])) OR (Pandemic, COVID-19[Title/Abstract]) |
| #10 | #8 OR #9 |
| #11 | #1 AND #4 AND #7 AND #10 |

random assignment concealment, blind method, incomplete result data, selective reporting bias, and other bias). The bias risk will include the risk of high, unclear, or low bias [37]. If there are any differences in the assessment, a third researcher will be consulted to resolve the dispute.

## Data synthesis and statistical analysis

Review Manager 5.3 software and Stata 14.0 software will be used for quantitative data analysis, including overall forest mapping, heterogeneity analysis, subgroup analysis, sensitivity analysis, and funnel mapping. For the diverse insomnia assessment tools, the standardized mean difference (SMD) and its 95% confidence interval (*CI*) will be used to describe the size of the effect. On the contrary, the mean difference (MD) and 95% *CI* will be used to represent the size of the effect.

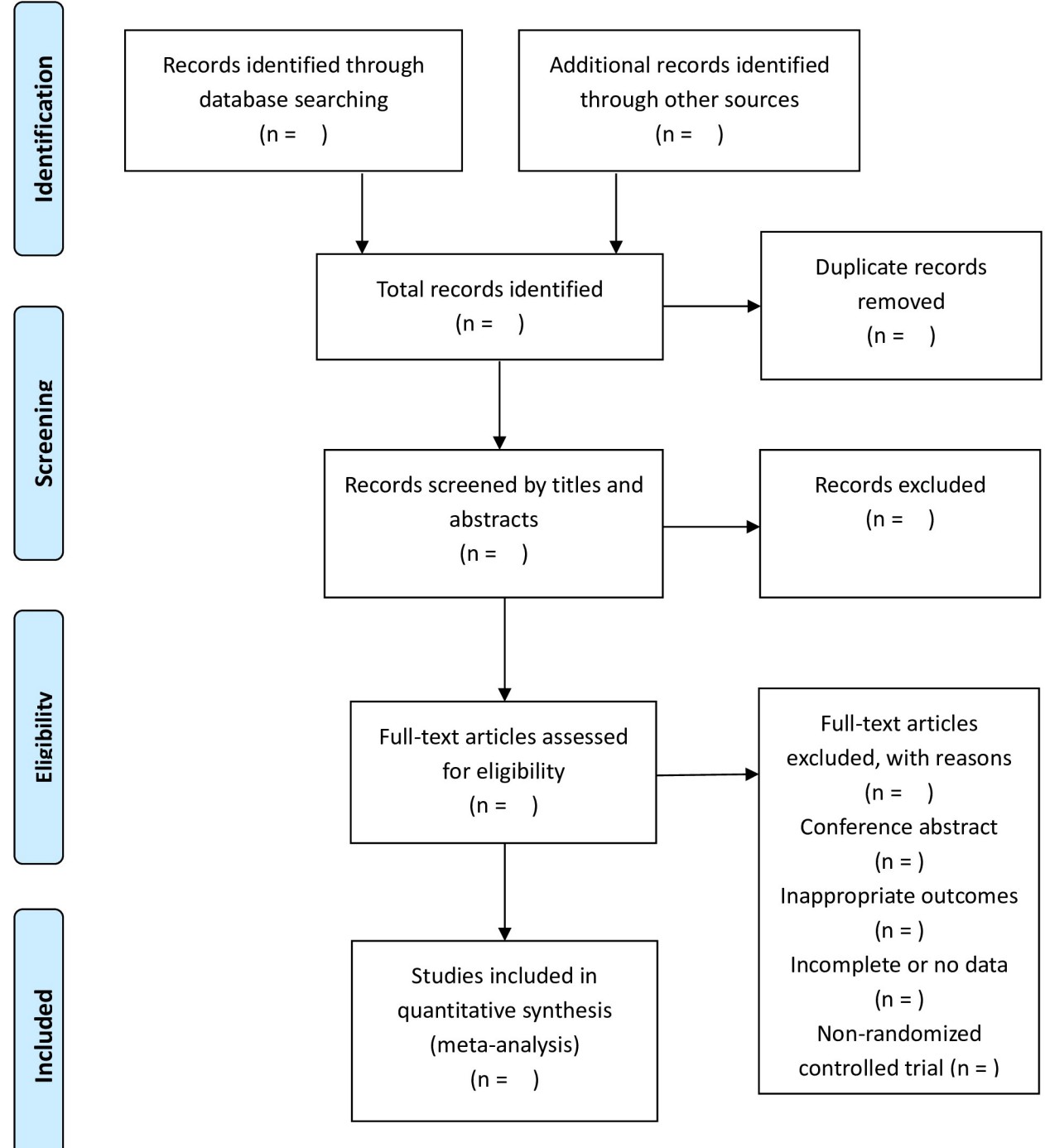

**Fig 1. Flow diagram of study selection process.**

## Assessment of heterogeneity

The heterogeneity assessment will be conducted using the Rev Man 5.3 software. The Chi-Squared test and $I^2$ will be adopted to examine the statistical heterogeneity of the included

studies. The random-effect model ($P \leq 0.1$ and $I^2 > 50\%$) or fixed-effect model ($P > 0.1$ and $I^2 \leq 50\%$) will be selected according to the values of $P$ and $I^2$.

## Subgroup analysis

When heterogeneity is found in the results, subgroup analysis will be performed on the basis of ensuring sufficient data. The criteria will include characteristics (such as country or sample source), study quality, type of control intervention, and duration. In addition, we will analyze any other subgroups reported in the literature.

## Assessment of reporting bias

Publication bias will be assessed by visual inspection of the funnel plot asymmetry and Egger's regression test [38].

## Sensitivity analysis

The robustness of analysis results will be explored by sensitivity analysis. Sensitivity analysis will be performed by excluding each uncertain factor to determine the sensitive factor exerting an important impact on the overall results.

## Quality of evidence

The researchers will assess the quality of evidence for the entire study in accordance with the Grades of Recommendation, Assessment, Development, and Evaluation system established by the WHO and international organizations. The GRADE system will rate the quality of evidence as "high," "medium," "low," and "lowest" [39].

# Discussion

Insomnia seriously impairs the quality of life of COVID-19 patients in convalescence and causes severe mental disorders such as anxiety, depression, and fear [40]. Therefore, the use of scientific psychological intervention to treat insomnia is of great practical significance. Previous studies have shown that Internet-delivered cognitive behavioral therapy can make insomnia patients achieve lasting sleep improvement over a period of time with fewer side effects [41]. Based on the existing evidence, we reasonably speculate that cognitive behavioral therapy can also relieve the insomnia of COVID-19 patients in convalescence.

Due to the impact of methodological quality, information bias, and adequacy of results reporting of the included literature, the therapeutic effect of Internet-delivered cognitive behavioral therapy on insomnia in COVID-19 patients in convalescence may be affected. Therefore, we need to conduct large-scale and high-quality research in the future to obtain more accurate results.

# Supporting information

**S1 Checklist. PRISMA-P 2009 checklist.**
(DOC)

**S1 Fig. Flow diagram of study selection process.**
(DOCX)

## Author Contributions

**Conceptualization:** Yan Chen, Rui Li, Zihan Jiang, Shuya Wang.

**Data curation:** Shuya Wang.

**Formal analysis:** Yan Chen, Shuya Wang.

**Funding acquisition:** Bo Li.

**Methodology:** Xiang Zhou.

**Software:** Xiang Zhou.

**Supervision:** Lina Wang.

**Writing – review & editing:** Yan Chen, Jie Liu, Zihan Jiang.

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
