## [Decision Letter · Decision Letter 0]

3 Mar 2022

PONE-D-21-40745Effect of internet-delivered cognitive behavioral therapy on insomnia in convalescent patients with COVID-19: protocol for a systematic review and meta-analysisPLOS ONE

Thank you for submitting your manuscript to PLOS ONE. After careful consideration, we feel that it has merit but does not fully meet PLOS ONE’s publication criteria as it currently stands. Therefore, we invite you to submit a revised version of the manuscript that addresses the points raised during the review process.

We look forward to receiving your revised manuscript.

Kind regards,

Luigi Lavorgna

Academic Editor

PLOS ONE

Journal Requirements:

4. We note that this manuscript is a systematic review or meta-analysis; our author guidelines therefore require that you use PRISMA guidance to help improve reporting quality of this type of study. Please upload copies of the completed PRISMA checklist as Supporting Information with a file name “PRISMA checklist”.

Reviewers' comments:

Reviewer's Responses to Questions

**Comments to the Author**

1. Does the manuscript provide a valid rationale for the proposed study, with clearly identified and justified research questions?

Reviewer #1: Partly

2. Is the protocol technically sound and planned in a manner that will lead to a meaningful outcome and allow testing the stated hypotheses?

Reviewer #1: Yes

3. Is the methodology feasible and described in sufficient detail to allow the work to be replicable?

Reviewer #1: Yes

4. Have the authors described where all data underlying the findings will be made available when the study is complete?

Reviewer #1: Yes

5. Is the manuscript presented in an intelligible fashion and written in standard English?

Reviewer #1: No

6. Review Comments to the Author

You may also provide optional suggestions and comments to authors that they might find helpful in planning their study.

Reviewer #1: The article by Chen et al. presented the study design of a metanalysis and systematic research to evaluate the effects of internet-delivered cognitive behavioral therapy on insomnia in COVID-19 patients in convalescence. Methods are sounds, however there are some minor concerns:

In the abstract section:

1) “In addition. I2 and Cochrane will be used for heterogeneity assessment.” I don’ understand the full stop just after “In addition”. AMEND

In the introduction section:

2) “and despair [9, 10], Even” comma or full stop? AMEND

3) “Insomnia is defined as the subjective perception of difficulties in falling asleep, maintaining asleep, and early morning awakening at least three times per week for at least 3 months[13].” Imsonnia not is a subjective perception and the article that is cited does not define it as a subjective perception. Please correct

4) “Till now, Previous studie” comma or full stop? AMEND

5) “But a recent study found that Internet cognitive”. Use “However” instead of “But”

6) There are many examples of internet-delivered therapy for neurological diseases and a recent review of the literature thoroughly presented them. I suggest to briefly discuss it in the introduction section (PMID: 34018047).

The article could improve with a revision by an english native speaker.

7. PLOS authors have the option to publish the peer review history of their article (what does this mean?). If published, this will include your full peer review and any attached files.

Reviewer #1: No

---

## [Author Response · Author response to Decision Letter 0]

24 Mar 2022

Dear Editor,

We would like to resubmit the revised manuscript entitled “Effect of Internet-delivered cognitive behavioral therapy on insomnia in convalescent patients with COVID-19: protocol for a systematic review and meta-analysis (PONE-D-21-40745)” for consideration by PLOS ONE. We would like to thank the reviewers for thoroughly reviewing our manuscript and making many thoughtful comments. We were very pleased to see that reviewers recognized the novelty and potential significance of our work. We have added significant new contents, described in detail below, and revised the manuscript to address reviewers’ comments. In the Revised Manuscript with Track Changes, the contents we added are marked in green and the contents we deleted are marked in red. Here are our point-by-point responses:

Yan Chen

Institute of Chronic Disease Risks Assessment, School of nursing and health,

Henan University

Kaifeng, Henan, China

E-mail address: chenyan8251@126.com

March 23, 2022

Journal Requirements:

Please ensure that your manuscript meets PLOS ONE's style requirements, including those for file naming. The PLOS ONE style templates can be found at https://journals.plos.org/plosone/s/file?id=wjVg/PLOSOne_formatting_sample_main_body.pdf and https://journals.plos.org/plosone/s/file?id=ba62/PLOSOne_formatting_sample_title_authors_affiliations.pdf

Answer: 

We have revised our manuscript to meet meets PLOS ONE's style requirements.

Answer: 

All relevant data from this study will be made available upon study completion.

Answer: 

Thank you for your comments. We have moved the ethics statement to the Methods section of the manuscript as required. Please review it again.

4. We note that this manuscript is a systematic review or meta-analysis; our author guidelines therefore require that you use PRISMA guidance to help improve reporting quality of this type of study. Please upload copies of the completed PRISMA checklist as Supporting Information with a file name “PRISMA checklist”.

Answer: 

We have used PRISMA guidelines to help improve the reporting quality of this study. At the same time, we have uploaded a copy of the completed PRISMA checklist as supporting information.

Answer: 

We have carefully examined the reference list to ensure its completeness and correctness.

Comments to the Author

Number 1:

1. Does the manuscript provide a valid rationale for the proposed study, with clearly identified and justified research questions?

Reviewer #1: Partly

Answer: 

Thank you for your comments.

Number 2:

2. Is the protocol technically sound and planned in a manner that will lead to a meaningful outcome and allow testing the stated hypotheses?

Reviewer #1: Yes

Answer: 

Thank you for your comments.

Number 3:

3. Is the methodology feasible and described in sufficient detail to allow the work to be replicable?

Reviewer #1: Yes

Answer: 

Thank you for your comments.

Number 4:

4. Have the authors described where all data underlying the findings will be made available when the study is complete?

Reviewer #1: Yes

Answer: 

Thank you for your comments.

Number 5:

5. Is the manuscript presented in an intelligible fashion and written in standard English?

Reviewer #1: No

Answer: Thank you very much for your valuable comments. We have carefully polished the language of the manuscript to make it readable and standard.

Number 6:

6. Review Comments to the Author

You may also provide optional suggestions and comments to authors that they might find helpful in planning their study.

Answer: 

Thank you.

Number 7:

Reviewer #1: The article by Chen et al. presented the study design of a metanalysis and systematic research to evaluate the effects of internet-delivered cognitive behavioral therapy on insomnia in COVID-19 patients in convalescence. Methods are sounds, however there are some minor concerns:

In the abstract section:

1) “In addition. I2 and Cochrane will be used for heterogeneity assessment.” I don’ understand the full stop just after “In addition”. AMEND

Answer: 

Thank you very much for your careful review. We have corrected the writing errors caused by our carelessness.

In the introduction section:

2) “and despair [9, 10], Even” comma or full stop? AMEND.

Answer: 

We have changed "comma" to "full stop" here.

3) “Insomnia is defined as the subjective perception of difficulties in falling asleep, maintaining asleep, and early morning awakening at least three times per week for at least 3 months[13].” Imsonnia not is a subjective perception and the article that is cited does not define it as a subjective perception. Please correct.

Answer: 

Thank you for your valuable comments. We have carefully checked the quoted articles and made the following changes to the definition of insomnia:

Insomnia is defined as difficulties in falling asleep, maintaining asleep, or waking up too early at least three times per week for at least 3 months [13] and is one of the most common problems experienced by convalescent patients with COVID-19 [14].

4) “Till now, Previous studie” comma or full stop? AMEND.

Answer: 

Thank you for your careful review. It is correct to use comma here, and we have changed the description of this sentence. Please review it again.

5) “But a recent study found that Internet cognitive”. Use “However” instead of “But”.

Answer: 

Thank you for your suggestion. We have replaced "But" with "However".

6) There are many examples of internet-delivered therapy for neurological diseases and a recent review of the literature thoroughly presented them. I suggest to briefly discuss it in the introduction section (PMID: 34018047).

Answer: 

Thank you for your constructive suggestions. We refer to a recent review and add the following contents to make the manuscript more comprehensive:

As a kind of digital therapy, Internet-delivered cognitive behavioral therapy provides patients with evidence-based treatment interventions, which are driven by software to prevent, manage, or treat medical disorders or diseases [31]. It can be used independently or in concert with medications, devices, or other therapies [31]. Internet-delivered cognitive behavioral therapy, as a promising method to solve psychotherapy obstacles [32], can overcome several limits related to traditional clinical practice [33].

The article could improve with a revision by an english native speaker.

Answer: 

We have carefully polished the language of the manuscript to make it more professional and more suitable for publication.

7. PLOS authors have the option to publish the peer review history of their article (what does this mean?). If published, this will include your full peer review and any attached files.

Do you want your identity to be public for this peer review? For information about this choice, including consent withdrawal, please see our Privacy Policy.

Reviewer #1: No

Thank you for your consideration of our manuscript.

Yours sincerely,

Yan Chen

---

## [Editor Report · Decision Letter 1]

4 Apr 2022

PONE-D-21-40745R1Effect of Internet-delivered cognitive behavioral therapy on insomnia in convalescent patients with COVID-19: protocol for a systematic review and meta-analysisPLOS ONE

Dear Dr. chen,

Thank you for submitting your manuscript to PLOS ONE. After careful consideration, we feel that it has merit but does not fully meet PLOS ONE’s publication criteria as it currently stands. Therefore, we invite you to submit a revised version of the manuscript that addresses the points raised during the review process.

The manuscript has been revised by one reviewer. Few inconsistency were found, in particular:

(1) the date of the search run was the 30th of July in the manuscript, while the the PROPSERO entry was the 11th of December. Please, can you clarify when the search was run?

(2) Please, in the Methods section, can you clarify whether the data have already been collected and analyzed?

(3) Ideally, Study Protocols should be written in the present tense. We noted that the tense used in the manuscript is the past tense. Could you please correct accordingly?

(4) We noted significant overlap between your work and the article titled ". Efficacy and safety evaluation of bright light therapy in patients with post-stroke insomnia: a protocol of systematic review and meta-analysis" (Lei H et al. Medicine. 2021;100:50(e27937)). Please, can you comment on this?

Please, revise the manuscript to carefully address all the concerns raised.

We look forward to receiving your revised manuscript.

Kind regards,

Luigi Lavorgna

Academic Editor

PLOS ONE
---

## [Author Response · Author response to Decision Letter 1]

6 Apr 2022

Dear Editor,

We would like to resubmit the revised manuscript entitled “Effect of Internet-delivered cognitive behavioral therapy on insomnia in convalescent patients with COVID-19: protocol for a systematic review and meta-analysis (PONE-D-21-40745R1)” for consideration by PLOS ONE. We would like to thank the reviewers for thoroughly reviewing our manuscript and making many thoughtful comments. We were very pleased to see that reviewers recognized the novelty and potential significance of our work. We have added significant new contents, described in detail below, and revised the manuscript to address reviewers’ comments. In the Revised Manuscript with Track Changes, the contents we added are marked in green and the contents we deleted are marked in red. Here are our point-by-point responses:

Yan Chen

Institute of Chronic Disease Risks Assessment, School of nursing and health,

Henan University

Kaifeng, Henan, China

E-mail address: chenyan8251@126.com

April 6, 2022

Journal Requirements: 

(1) The date of the search run was the 30th of July in the manuscript, while the PROPSERO entry was the 11th of December. Please, can you clarify when the search was run?

Answer: Thank you very much for your careful review. The search run in the registration protocol is the 30th of July. However, given the prevalence of COVID-19 epidemic, we consider that there may be more published articles that meet the criteria of the study. To expand the search scope, the search run in the manuscript is the 11th of December. Combined with the your comments, we have changed the retrieval time to the 30th of July in the manuscript to make it consistent with the research protocol.

(2) Please, in the Methods section, can you clarify whether the data have already been collected and analyzed?

Answer: Thank you for your valuable comments. Till now, the data have already been collected and analyzed. We have clarified it in the Methods section.

(3) Ideally, Study Protocols should be written in the present tense. We noted that the tense used in the manuscript is the past tense. Could you please correct accordingly?

Answer: Thank you very much for your valuable comments. We have carefully modified the tense used in the manuscript.

(4) We noted significant overlap between your work and the article titled ". Efficacy and safety evaluation of bright light therapy in patients with post-stroke insomnia: a protocol of systematic review and meta-analysis" (Lei H et al. Medicine. 2021;100:50(e27937)). Please, can you comment on this?

Answer: Thanks for your careful review. As you pointed out, the study entitled "Efficacy and safety evaluation of bright light therapy in patients with post-stroke insomnia: a protocol of systematic review and meta-analysis (PMID: 34918641) has some similarities in the selection of outcome indicators with our study. However, after careful reading, we find great differences in the design of the two studies. Firstly, the research objects are different. The research objects of our study are COVID-19 patients in rehabilitation, while the above study (PMID:34918641) focuses on the insomnia population with stroke. Although the two studies all use insomnia as an outcome indicator, insomnia in stroke patients may be caused by nervous system damage, and our study is aimed at insomnia in COVID-19 rehabilitation patients. Secondly, the intervention methods are different. Our study uses Internet-delivered cognitive behavioral therapy. Cognitive behavioral therapy, as an evidence-based option, has been found to effectively improve sleep outcomes [1, 2] and its therapeutic effect is equivalent to or better than that of drug therapy [3]. However, the above study (PMID:34918641) selects bright light therapy (Typically, patients are instructed to be exposed to bright light at a constant distance every day[4]).

References

[1] Trauer JM, Qian MY, Doyle JS, Rajaratnam SM, Cunnington D. Cognitive Behavioral Therapy for Chronic Insomnia: A Systematic Review and Meta-analysis. Ann Intern Med. 2015 4; 163(3):191-204. https://doi.org/10.7326/M14-2841 PMID: 26054060

[2] Irwin MR, Cole JC, Nicassio PM. Comparative meta-analysis of behavioral interventions for insomnia and their efficacy in middle-aged adults and in older adults 55+ years of age. Health Psychol. 2006; 25(1):3-14. https://doi.org/10.1037/0278-6133.25.1.3 PMID: 16448292

[3] Smith MT, Perlis ML, Park A, Smith MS, Pennington J, Giles DE, Buysse DJ. Comparative meta-analysis of pharmacotherapy and behavior therapy for persistent insomnia. Am J Psychiatry. 2002; 159(1):5-11. https://doi.org/10.1176/appi.ajp.159.1.5 PMID: 11772681

[4] van Maanen A, Meijer AM, van der Heijden KB, Oort FJ. The effects of light therapy on sleep problems: A systematic review and meta-analysis. Sleep Med Rev. 2016; 29:52-62. http://doi.org/ 10.1016/j.smrv. PMID: 26606319

---

## [Editor Report · Decision Letter 2]

11 Apr 2022

PONE-D-21-40745R2Effect of Internet-delivered cognitive behavioral therapy on insomnia in convalescent patients with COVID-19: protocol for a systematic review and meta-analysisPLOS ONE

Thank you for submitting your manuscript to PLOS ONE. After careful consideration, we feel that it has merit but does not fully meet PLOS ONE’s publication criteria as it currently stands. Therefore, we invite you to submit a revised version of the manuscript that addresses the points raised during the review process.

We look forward to receiving your revised manuscript.

Kind regards,

Luigi Lavorgna

Academic Editor

PLOS ONE

Journal Requirements:

Additional Editor Comments (if provided):

"From an evaluation of your revision, we noted that there still are concerning issues that need to be resolved. In particular:

1) we noted that the data analysis for this study in now completed, as you reported in the Methods section. Study Protocol are articles type for which data collection and analysis cannot be completed at the moment of submission. Please, remove the sentence from the Methods.

2) Please, edit your manuscript with the present or future tense in order to make it conform to the Study Protocol criteria. Study protocol present the proposal of a study, and the manuscript should read as something you are proposing to do.

3) The search date between the manuscript and the PROSPERO entry are still inconsistent. As you extended the search strategy to the 11th of December, please report this into the main text of the manuscript.

4) We appreciate your response on the differences between your study and Lei H et al. Medicine. 2021;100:50(e27937). However, we feel you did not provide sufficient explanation on why the search strategies are similar. Please, clarify this point."
---

## [Author Response · Author response to Decision Letter 2]

16 Apr 2022

Dear Editor,

We would like to resubmit the revised manuscript entitled “Effect of Internet-delivered cognitive behavioral therapy on insomnia in convalescent patients with COVID-19: protocol for a systematic review and meta-analysis (PONE-D-21-40745R1)” for consideration by PLOS ONE. We would like to thank the reviewers for thoroughly reviewing our manuscript and making many thoughtful comments. We were very pleased to see that reviewers recognized the novelty and potential significance of our work. We have added significant new contents, described in detail below, and revised the manuscript to address reviewers’ comments. In the Revised Manuscript with Track Changes, the contents we added are marked in green and the contents we deleted are marked in red. Here are our point-by-point responses:

Yan Chen

Institute of Chronic Disease Risks Assessment, School of nursing and health,

Henan University

Kaifeng, Henan, China

E-mail address: chenyan8251@126.com

April 17, 2022

Journal Requirements: 

1) we noted that the data analysis for this study in now completed, as you reported in the Methods section. Study Protocol are articles type for which data collection and analysis cannot be completed at the moment of submission. Please, remove the sentence from the Methods.

Answer: Thank you for your valuable comments. We have deleted the sentence "Till now, the data have already been collected and analyzed." and further clarified it in the Methods.

2) Please, edit your manuscript with the present or future tense in order to make it conform to the Study Protocol criteria. Study protocol present the proposal of a study, and the manuscript should read as something you are proposing to do.

Answer: Thank you very much for your valuable comments. We have carefully modified the tense used in the manuscript.

3) The search date between the manuscript and the PROSPERO entry are still inconsistent. As you extended the search strategy to the 11th of December, please report this into the main text of the manuscript.

Answer: Thank you very much for your careful review. The search run in the registration protocol is the 30th of July. However, given the prevalence of COVID-19 epidemic, we consider that there may be more published articles that meet the criteria of the study. Hence, the search run in the manuscript is the 11th of December so as to expand the search scope. In consideration of your comments, we have clarified it in the main text of the manuscript.

4) We appreciate your response on the differences between your study and Lei H et al. Medicine. 2021;100:50(e27937). However, we feel you did not provide sufficient explanation on why the search strategies are similar. Please, clarify this point."

Answer: Thanks for your careful review. As you pointed out, the study entitled "Efficacy and safety evaluation of bright light therapy in patients with post-stroke insomnia: a protocol of systematic review and meta-analysis (PMID: 34918641) has some similarities with our study in search strategies.

After careful review, we found that the similarity is mainly due to two reasons. Firstly, this study (PMID: 34918641) is different from our study in the research object, but it is the same in the analysis of the outcome index, namely insomnia. Secondly, both the study (PMID:34918641) and our study use subject words [Sleep Initiation and Maintenance Disorders[Mesh]] combined with entry terms [(Disorders of Initiating and Maintaining Sleep) OR (Disorders of Initiating and Maintaining Sleep) OR (Early Awakening) OR (Awakening, Early) OR (Nonorganic Insomnia) OR (Insomnia, Nonorganic) OR (Primary Insomnia) OR (Insomnia, Primary) OR (Transient Insomnia) OR(Insomnia, Transient) OR (Rebound Insomnia) OR (Insomnia, Rebound) OR (Secondary Insomnia) OR (Insomnia, Secondary) OR (Sleep Initiation Dysfunction) OR (Dysfunction, Sleep Initiation) OR (Dysfunctions, Sleep Initiation) OR (Sleep Initiation Dysfunctions) OR (Sleeplessness) OR (Insomnia Disorder) OR (Insomnia Disorders) OR (Insomnia) OR (Insomnias) OR (Chronic Insomnia) OR (Insomnia, Chronic) OR (Psychophysiological Insomnia) OR (Insomnia, Psychophysiological)] to retrieve the PubMed. This scientific approach can help to obtain more comprehensive search results. Therefore, due to the same outcome index and search method, the two studies are similar in search strategies.

Thank you for your consideration of our manuscript.

Yours sincerely,

Yan Chen

---

## [Editor Report · Decision Letter 3]

30 May 2022

Effect of Internet-delivered cognitive behavioral therapy on insomnia in convalescent patients with COVID-19: protocol for a systematic review and meta-analysis

PONE-D-21-40745R3

We’re pleased to inform you that your manuscript has been judged scientifically suitable for publication and will be formally accepted for publication once it meets all outstanding technical requirements.

Kind regards,

Luigi Lavorgna

Academic Editor

PLOS ONE
---

## [Editor Report · Acceptance letter]

2 Jun 2022

PONE-D-21-40745R3 

Effect of Internet-delivered cognitive behavioral therapy on insomnia in convalescent patients with COVID-19: protocol for a systematic review and meta-analysis 

Dear Dr. Chen:

I'm pleased to inform you that your manuscript has been deemed suitable for publication in PLOS ONE. Congratulations! Your manuscript is now with our production department. 

Kind regards, 

on behalf of

Dr. Luigi Lavorgna 

Academic Editor

PLOS ONE